



# Long-term change in the contributions of various source regions to surface ozone over Japan

**Tatsuya Nagashima[1], Kengo Sudo[2,3], Hajime Akimoto[1], Junichi Kurokawa[4], and Toshimasa Ohara[1]**

[1]National Institute for Environmental Studies, Tsukuba, Japan

[2]Graduate School of Environmental Studies, Nagoya University, Nagoya, Japan

[3]Frontier Research Center for Global Change, Yokohama, Japan

[4]Asia Center for Air Pollution Research, Niigata, Japan

*Correspondence to:* T. Nagashima (nagashima.tatsuya@nies.go.jp)

**Abstract**

The relative contributions of various source regions to the long-term (1980–2005) increasing trend in surface ozone ($O_3$) over Japan were estimated by a series of tracer-tagging simulations using a global chemical transport model. The model well simulated the observed increasing trend of surface $O_3$ including its seasonal variation and geographical features in Japan and demonstrated the relative roles of different source regions in forming this trend. Most of the simulated increasing trend of surface $O_3$ over Japan (~97 %) was explained as the sum of trends in contributions of different regions to photochemical $O_3$ production. The increasing trend in $O_3$ produced in China accounted for 36 % of the total increasing trend and those in the other northeast Asian regions (the Korean Peninsula, coastal regions in East Asia, and Japan) each accounted for about 12–15 %. Furthermore, the contributions of $O_3$ created in the entire free troposphere and in West, South, and Southeast Asian regions also increased; and their increasing trends accounted for 16 and 7 % of the total trend, respectively. The impact of interannual variations in climate, in methane concentration, and in emission of $O_3$ precursors from different source regions on the relative contributions of $O_3$ created in each region estimated above was also investigated. The variation of climate and the increase in methane concentration together caused the increase of photochemical $O_3$ production in several regions, and represented about 19 % of the total increasing trend of surface $O_3$ over Japan. The increase in emission of $O_3$ precursors in China caused an increase of photochemical $O_3$ production not only in China itself but also in the other northeast Asian regions and accounted for about 46 % of the total increase in surface $O_3$ over Japan. Similarly, the relative impact of $O_3$ precursor emission changes in the Korean Peninsula and Japan were estimated as about 16 and 4 % of the total increasing trend, respectively. The $O_3$ precursor emission change in regions other than northeast Asia caused increases in surface $O_3$ over Japan mainly through increasing photochemical $O_3$ production in West, South, and Southeast Asia and the free troposphere, and accounted for about 16 % of the total.



## 1 Introduction

Tropospheric ozone (O₃) plays multiple roles in the atmosphere. O₃ itself is an oxidant and photodissociates to generate the hydroxyl radical which strongly oxidizes many atmospheric compounds including various air pollutants and thus removes them from the atmosphere. In contrast, high levels of O₃ are a major air pollutant due to adverse effects on human health, natural vegetation, and agricultural produce (Wang and Mauzerall, 2004; Mauzerall et al., 2005; US EPA, 2006; Silva et al., 2013). Moreover, tropospheric O₃ is a major greenhouse gas in the atmosphere, and reduction of its amount was recently recognized as an effective measure to mitigate near-term climate change (UNEP and WMO, 2011; Shindell et al., 2012). Therefore, the spatial and temporal variations in tropospheric O₃ have been always a matter of scientific and public concern.

An increasing trend in tropospheric O₃ concentration has been observed during recent decades at many locations in East Asia including Taiwan (Chou et al., 2006; Chang and Lee; 2007; Li et al., 2010; Lin et al., 2010), mainland China (Lu and Wang, 2006; Ding et al., 2008; Xu et al., 2008; Wang et al., 2009; Zhang et al., 2014), and South Korea (Susaya et al., 2013; Lee et al., 2014; Seo et al., 2014). The increase rates of O₃ in those East Asian regions significantly vary depending on location and season in the range of about 0.3–3 ppbv/yr; however, the increases are generally larger than the trends in tropospheric O₃ for other regions in the world (Cooper et al., 2014). Japan is no exception, with an increasing trend found in various observations of O₃ over the past approximately 40 years. Routine ozonesonde measurements since 1970 at three Japanese sites of Sapporo (43° N), Tsukuba (36° N), and Kagoshima (32° N) showed an increasing trend of O₃ concentration in the lowermost troposphere up to about 1990 and relatively stable thereafter, with largest increase near the ground and discernible about 300 hPa height and below (Logan et al., 1999; Oltmans et al., 2006). With an air mass classification method based on backward air trajectories, Naja and Akimoto (2004) showed that a significant amount of the air masses reaching these ozonesonde sites in Japan spend substantial time over polluted regions in East Asia. The O₃ levels in these regionally polluted air masses increased from the 1970s to the 1990s, mainly due to large increases in nitrogen oxide (NOₓ = NO + NO₂) emissions over China in the 1990s. Oltmans et al. (2013) analyzed a rather short period of data (1991–2010) obtained at the Ryori (39° N) surface site in north-eastern Japan and showed an increase into the mid-1990s followed by relatively little change. Other ground-based observations at a mountain site (Mt. Happo; 43° N, 1850 m asl) and three sites in the marine boundary layer along the west coast of Japan [Rishiri (45° N), Tappi (41° N), and Sado (38° N)], where few sources of pollutants exist nearby, obtained under the monitoring network of EANET (the Acid Deposition Monitoring Network in East Asia) also showed increasing trends of O₃ concentrations at least until the mid-2000s (Tanimoto, 2009; Tanimoto et al., 2009; Parrish et al., 2012).

In addition, analysis of long-term observations by the ambient air quality monitoring network mainly established in urban–suburban regions in Japan also showed continuous increases of surface O₃ from the mid-1980s until the present (Ohara and Sakata, 2003; Ohara et al., 2008; Kurokawa et al., 2009; MOE Japan, 2013; Wakamatsu et al., 2013; Akimoto et al., 2015). However, simultaneous observations of O₃ precursors such as NOₓ and non-methane hydrocarbons (NMHCs) by this monitoring network revealed their decreasing trends in the same period (MOE Japan, 2013), which seemed inconsistent with the increasing trend of O₃ over Japan. These observed features of O₃-related atmospheric species in Japan suggest that there should be an influence of transboundary transport from outside of Japan on the recent increasing trend in O₃. The influence of transboundary transport on surface O₃ in East Asia was examined in several studies (Sudo and Akimoto, 2007; Li et al., 2008; Nagashima et



al., 2010; Wang et al., 2011). Nagashima et al. (2010) demonstrated that the $O_3$ transported
from outside of Japan accounted for more than 70 % of surface $O_3$ over Japan in the cold
season (October–March) during 2000–2005, and most was attributable to $O_3$ from distant
sources outside East Asia and from the stratosphere. In the warm season (April–September),
the contribution of domestically created $O_3$ in Japan to surface $O_3$ over Japan increased
significantly (about 20–40 %), the short range intra-regional transport of $O_3$ from other parts
of East Asia still contributed about 25 %, and long range inter-regional transport of $O_3$ from
outside East Asia and the stratosphere particularly in spring could account for about half of
surface $O_3$ over Japan.
Therefore, the influence of $O_3$ from source regions outside and inside East Asia and the
stratosphere should be considered to explain the cause of the increasing trend in surface $O_3$
over Japan. The rapid increase in $O_3$ precursor emissions in East Asia in recent decades
(Ohara et al., 2007; Kurokawa et al., 2013) was demonstrated as a major cause of the
increasing trend of springtime $O_3$ over Japan by comparing regional chemical transport model
(CTM) simulations of recent decades with and without the East Asian $O_3$ precursor emission
increases during the period (Kurokawa et al., 2009; Tanimoto et al., 2009). However, they
only showed the springtime $O_3$ case and it was unclear whether the relationship held in other
seasons. Moreover, the relative contributions of individual countries or regions in East Asia
have not been well examined, particularly concerning increased surface $O_3$ over Japan.
Here, we investigated the cause of the continuous increase in surface $O_3$ over Japan reported
in the above literature, focusing on the relative contributions of various source regions over
the globe, particularly the contributions of individual regions in East Asia, with a long-term
simulation of a global CTM using the tagged tracer method. Using the same model and
method, Nagashima et al. (2010) showed such relative contributions of regions inside and
outside East Asia on surface $O_3$ over Japan as average values for the early 2000s. The current
study investigated the temporal evolution of the relative contributions of each region for the
26 years of 1980–2005.

## 2   Methods

### 2.1   Model description

In this study, we employed a chemistry climate model (CCM), CHASER (Sudo et al., 2002),
developed for the atmospheric chemistry research in the troposphere. The basic setting of the
model was almost identical to that used by Nagashima et al. (2010). However, the horizontal
resolution was modified from T63 (about 1.9° by 1.9° grid spacing in longitude and latitude)
to T42 (about 2.8° by 2.8°), because longer simulation period was necessary than in the
previous study, and so the cost of computation was reduced in the present study by selecting
lower horizontal resolution. There were 32 vertical layers with the top layer set at
approximately 40 km altitude. A detailed tropospheric photochemistry consisted of 113
chemical reactions and 27 photodissociation involving $O_3$, $HO_x$, $NO_x$, methane ($CH_4$), CO and
NMHCs calculated the temporal evolution in the concentrations of 53 chemical species. The
gas and liquid phase oxidation of sulfur dioxide ($SO_2$) and dimethyl sulfide to form the sulfate
aerosol was also included in the model. The concentrations of $O_3$ and some nitrogen
compounds ($NO_x$, $HNO_3$, and $N_2O_5$) above the tropopause that should affect tropospheric
chemistry were assimilated into the monthly mean output data of stratospheric CCM, because
the version of CHASER used was unable to calculate several chemical processes, such as
halogen-related chemical reactions, which are indispensable for realistic representation of





such chemical compounds in the stratosphere. For the transport of chemical species, a semi-
Lagrangian advective transport scheme (Lin and Rood, 1996; van Leer, 1997) and vertical
convective transport associated with cumulus convection process were considered. The model
also included dry and wet deposition of chemical species.
In this study, we conducted tracer-tagging simulation by using two different setups (full-
chemistry and tracer-transport setups) of CHASER. The full-chemistry setup calculated the
actual temporal change in the concentration of chemical species through the abovementioned
chemical and physical processes and outputted the chemical production and loss tendencies of
$O_3$ and related species. Then, the tracer-transport setup used the outputted chemical
tendencies to calculate the temporal change in the concentration of hypothetical $O_3$ tracers. In
the following subsection, the calculation procedure is briefly described.

## 2.2  Outline of the numerical simulations

### 2.2.1 Forcings for long-term simulation

Long-term simulation was performed for the period 1980–2005. To drive the physical
properties of the model for this 26-year period, the temperature and horizontal wind velocities
in the model were assimilated into the National Center for Environmental Prediction/National
Center of Atmospheric Research (NCEP/NCAR) 6-hour reanalysis data (Kalnay et al., 1996)
of the corresponding year, and sea surface temperature and sea ice data of the Hadley Centre's
Sea Ice and Sea Surface Temperature (HadISST) data set (Rayner et al., 2003) were used in
the model.
The monthly mean stratospheric $O_3$ data of Akiyoshi et al. (2009) was used for the
assimilation above the tropopause for this period. These data were the output of a
stratospheric CCM simulation according to the hindcasting scenario for 1980–2004 (REF1
scenario) of the CCM validation activity (CCMVal) (Eyring et al., 2005), and included an
interannual variation (IAV) associated with the 11-year solar cycle and large declines after
1982 and 1991 due to the El Chichon and Pinatubo eruptions, respectively, in addition to a
continuous decreasing trend during the whole period. Although the simulated declines of
stratospheric $O_3$ due to the two large volcanic eruptions were somewhat overestimated, the
simulated IAVs in stratospheric $O_3$ reasonably well represented those observed with a total
ozone mapping spectrometer (TOMS) from satellites (Akiyoshi et al., 2009). Incidentally, the
stratospheric $O_3$ data of 2004 were used for 2005.
The long-term variation of the emissions of $O_3$ precursors ($NO_x$, CO, and NMHCs) and $SO_2$
were taken from multiple emission inventories. For anthropogenic emissions in Asia, the
Regional Emission inventory in ASia (REAS ver.1.2) (Ohara et al., 2007) was used for the
whole simulation period (1980–2005); the REAS emission data were available for each year
in the period. Kurokawa et al. (2009) used these emission data with a regional air quality
model representing well the interannual variability of surface $O_3$ over Japan for similar period
(1981–2005) to the present study. For anthropogenic emissions outside Asia, a combination of
three versions of EDGAR (Emission Database for Global Atmospheric Research) emission
data was used: EDGAR-HYDE (Van Aardenne et al., 2001) for 1980 and 1990; EDGAR v3.2
(Olivier and Berdowski, 2001) for 1990 and 1995; and EDGAR v3.2 Fast Track 2000
(FT2000) (Olivier and Berdowski, 2001) for 2000. Because several emission sectors
considered in EDGAR v3.2 were not considered in EDGAR-HYDE, the emissions for 1990
in EDGAR-HYDE were generally smaller than in EDGAR v3.2. Therefore, we used EDGAR





v3.2 data for 1990, and also scaled them to estimate emission data for 1980 rather than simply
using EDGAR-HYDE data for 1980. For that, we scaled EDGAR v3.2 data for 1990 so that
the ratio (r) of the difference between 1980 ($f_1$) and 1990 data ($f_2$) and their average in
EDGAR-HYDE [i.e., $r = (f_2 - f_1)/(f_1 + f_2)/2$] equaled the corresponding ratio (R) calculated
from 1990 data in EDGAR v3.2 ($F_2$) and 1980 data scaled from it ($F_1$) [i.e., $R = (F_2 - F_1)/(F_1 +$
$F_2)/2$]. We calculated $F_1$ from the known values of $f_1$, $f_2$, and $F_2$ using the equation $r = R$.
Since EDGAR emission data were not available for each year but for every 10 or 5 years in
the simulation period, the emissions for intermediate years were interpolated, and FT2000
data used for years after 2000. The vegetation fire emission data developed in the REanalysis
of the TROpospheric chemical composition over the past 40 years project (RETRO) (Schultz
et al., 2008) were used for $O_3$ precursor emissions from biomass burning for the whole land
area. RETRO data were available for each year until 2000 in the simulation period, and data
for 2000 were used for years after 2000. Historical transition of the atmospheric
concentrations of carbon dioxide, nitrous oxide ($N_2O$), and $CH_4$ were prescribed with those
used in Nozawa et al. (2005), which were somewhat old estimations of the historical
evolution in greenhouse gas concentrations, but not much different from recent estimations
such as for the Representative Concentration Pathways (RCPs) (Meinshausen et al., 2011).
The difference in the concentrations between both estimations were generally within a couple
of percent in the simulation period.
The linear trends of $NO_x$ and NMVOC annual emissions used in this study in the simulation
period of 1980–2005 are shown in Fig. 1. The long-term trends of emissions of both species
showed generally similar geographical features to each other; large decrease trends in central
Europe, Scandinavia, western Russia, and Kazakhstan, whereas there were widely spread
increasing emissions in West, South, Southeast, and East Asia, almost all Africa and Central
and South America except for inland Brazil. In North America, $NO_x$ emission generally
decreased in the simulation period except for the west coast and New England area of the
USA, but that of NMVOC mostly increased with a few patchy exceptions. The trends of $NO_x$
and NMVOC emissions mentioned above were mainly due to the change in anthropogenic
emissions, while the change in biomass burning emissions led to a discernible trend in several
regions such as inland Brazil and the south of Sahel.
The long-term evolution of annual emissions of $NO_x$ and NMVOC over several source areas
in the Northern Hemisphere is shown in Fig. 2. Because the emission data were the
combination of three different datasets outside Asia, there were somewhat discontinuous
changes at the joint years (1990 and 1995) in European and North American emissions. The
emissions of $NO_x$ and NMVOC over Europe had peaks around 1990 and generally decreased
afterward. Over North America, both species showed small long-term trends: slight decreases
in $NO_x$ and slight increases in NMVOC emissions. The emissions of both species over China
greatly increased during the whole period. The $NO_x$ emissions were about 4.0 times larger in
2005 than 1980 and correspondingly NMVOC was 2.5 times larger, which made emissions of
both species for China equal to or even surpassing those for Europe or North America in 2005.
The emissions of both species over the Korean Peninsula increased approximately 2.8 times
during this period. However, those over Japan showed no such increase: $NO_x$ emission
decreased until 1995 and thereafter remained stable, whereas NMVOC emissions went up
until 1995 and then slightly decreased.

### 221     2.2.2 Tracer tagging

We conducted a 26-year simulation using the full-chemistry setup of CHASER with all the
forcings mentioned above, followed by another 26-year simulation with the tracer-transport



setup of CHASER which calculated the concentration of hypothetical $O_3$ tracers, each tagged
with a particular region in the model domain. The procedure to tag a tracer with each region
in the second simulation was the same as used by Nagashima et al. (2010) and a brief
description follows. In the second simulation, the transport and dry deposition of each $O_3$
tracer were calculated same as in the first simulation, however the chemical development of
tracers was calculated using the chemical production (P) and loss frequencies (L) of the
extended odd oxygen family $[O_x = O_3 + O + O(^1D) + NO_2 + 2NO_3 + 3N_2O_5 + PANs + HNO_3$
+ other nitrates] calculated and archived in the first simulation. In the first simulation, 3D
fields of P and L were outputted every 6 hours. Each $O_3$ tracer could be lost chemically
everywhere in the model domain at the frequency of L, but could be chemically produced
only inside its tagged region. In the stratosphere over the tropopause defined by the lapse rate,
the concentration of $O_3$ tracer tagged with the stratosphere was assimilated into the same
stratospheric $O_3$ data as used in the first simulation, but the concentration of the tracers tagged
with the region in the troposphere were all set to zero. The calculated concentration of each
tagged $O_3$ tracer at a given location represents the contribution of $O_3$ produced in each source
region and transported to that location.
The horizontal and vertical separation of the model domain for the tracer tagging was also
the same as used by Nagashima et al. (2010). The troposphere in the model domain was
horizontally separated into 22 regions and each horizontal region was further separated
vertically between the free troposphere (FT) and the planetary boundary layer (PBL). The
stratosphere was considered one separate source region, that is, the model domain was
separated into 45 source regions. The 22 regions for horizontal separation are shown in Fig. 1
and each region was assigned a three-letter code (e.g., AMN for North America) which is
used in the following sections. For the vertical separation of the source regions in the
troposphere, the PBL was defined as the lowest six layers in the model (surface to about 750
hPa), based on the observed and modeled vertical profiles of $O_3$ production.
The long-term tracer-tagging simulation allowed estimation of the long-term variations in
contributions of each source region to the $O_3$ concentration at given receptor locations. This is
important information to explain the cause of the reported increasing trend in surface $O_3$ over
East Asia. However, it should be noted that the tracer-tagging simulation calculates the
amount of $O_3$ in a receptor location that was produced chemically in each source region from
$O_3$ precursors emitted both from the source region and adjacent source regions. Thus, the
contribution of a source region estimated in tracer-tagging simulation should not be fully
attributed to emissions of $O_3$ precursors in that source region. Emission sensitivity simulation
is another method of estimating the portion of $O_3$ fully attributable to a change in $O_3$ precursor
emissions in a source region, and takes the difference of simulated $O_3$ between two model
runs with and without perturbed $O_3$ precursor emissions in that source region. The resulting
estimations of source contributions by the two methods can differ; however, the differences
have not yet been well quantified. Li et al. (2008) reported that the difference between the two
methods could be as much as 30 % in source apportionment estimation for one location and
time (i.e., Mt. Tai in central eastern China in June 2006). Wang et al. (2011) found somewhat
larger differences in the contributions of China to domestic $O_3$ concentration between the two
methods for each month of the year, but no discussions were made for $O_3$ over Japan.
Nevertheless, we employed the tracer-tagging simulation to study the cause of reported long-
term change in surface $O_3$ over Japan mainly due to its computational efficiency. Thus, the
results should be carefully interpreted in terms of the difference between the source regions of
chemical $O_3$ production and those of $O_3$ precursor emissions. The computational efficiency
resulting from the tracer-tagging approach and relatively coarse horizontal resolution enabled



us to make several sensitivity simulations with the different combination of forcings for long-
term simulation. In the following sections, the simulation with the full set of long-term
forcings described above, hereinafter referred to as "standard" simulation, is initially analyzed.
This is then further interpreted using the results of sensitivity simulations; the specific settings
of sensitivity simulations are also described.

## 3 Results and discussion

### 3.1 Long-term evolution of surface $O_3$ over Japan

Nagashima et al. (2010) validated how well CHASER can reproduce the observed features
of surface $O_3$ concentrations by comparing the simulated surface $O_3$ concentrations with
observations taken during 2000–2005 at several sites mainly in rural areas in the Northern
Hemisphere, and CHASER successfully simulated the annual variation of surface $O_3$ in a
variety of regions. In this study, the horizontal resolution of the model differed from that used
in Nagashima et al. (2010); however, the model well represented the observed concentrations
and seasonal evolutions of surface $O_3$ (Fig. S1 and Table S1 in the Supplement). The surface
$O_3$ over Japan has been observed at ambient air quality monitoring stations since the early
1970s when severe air pollution occurred in industrial or urban areas. The monitoring data
have been compiled by the Atmospheric Environmental Regional Observation System
(AEROS). The number of stations increased since the launch of the system and, for the period
of simulation (1980–2005), about 1000 monitoring stations widely distributed throughout
Japan except in the southern islands could be used for validation of the model results. The
monitoring data of AEROS have been used to examine the long-term variation of surface $O_3$
over Japan in several studies and showed significant increasing trends (Ohara and Sakata,
2003; Ohara et al., 2008; Kurokawa et al., 2009; Akimoto et al., 2015). We validated the
simulated surface $O_3$ over Japan with the AEROS data in terms of the long-term variation in
the following.
For the validation, the monitoring sites selected had continuously observed the surface $O_3$
during the simulation period (1980–2005). To ensure continuity of sites, we selected
monitoring sites with annual mean surface $O_3$ available for every year in the simulation period.
The annual mean data at a monitoring site was calculated as the average of monthly means
when available for more than 9 months, the monthly mean was calculated from daily means
when available for over 19 days per month, and the daily mean was calculated from hourly
means when available for more than 19 hours per day. There were 339 sites, located mainly in
populated areas of Japan except in the northernmost island (Hokkaido) and southern islands
(Nansei Islands). We first calculated the annual mean surface $O_3$ from the observed hourly
data at each monitoring site as described above, and then the annual means of all sites were
averaged to calculate the observed annual mean surface $O_3$ over Japan. The simulated annual
mean surface $O_3$ over Japan was calculated as the average of annual means of the model grids,
which included the locations of monitoring sites selected for the validation. Therefore, the
model grids including Hokkaido or Nansei Islands were not used to calculate the simulated
annual mean. The temporal variations of observed and simulated annual mean surface $O_3$
anomalies during 1980–2005 averaged over Japan are shown in Fig. 3. During the period, the
observed annual mean surface $O_3$ over Japan showed a clear increasing trend with a linear
increase of about 2.70 ppbv/decade, which was significant at the 5 % risk level. The simulated
annual mean surface $O_3$ over Japan also showed a significant increasing trend with a rate of
about 2.58 ppbv/decade, which corresponded well to the observed increase in surface $O_3$ over





Japan. The value of the linear increasing trend and the observed features of IAVs in surface
$O_3$ over Japan – such as a rapid increase from the mid-1980s to the mid-1990s followed by a
stagnation of increase for about 7–8 years and a further increase in the past several years –
were reasonably well captured by the model.
The model also well represented the longitudinal differences in the long-term trend of
surface $O_3$ in Japan. Figure 4 shows the maps of linear trends of annual mean surface $O_3$
during 1980–2005 calculated from the model simulations and observations at AEROS
monitoring sites as selected for Fig. 3. The simulated annual mean surface $O_3$ showed an
increasing trend in the whole area including all of Japan and the Korean Peninsula (Fig. 4a).
The simulated increasing trend of annual mean surface $O_3$ well exceeded 2.0 ppbv/decade in
wide areas of Japan except for Hokkaido, and tended to be greater toward western Japan,
which is nearer to the Asian continent. However, the increasing trends of observed annual
mean surface $O_3$ at each monitoring site (Fig. 4b) differed greatly from each other even in
nearby sites, and there was no apparent longitudinal tendency in trends at individual
monitoring sites. However, we averaged the observed annual mean surface $O_3$ at individual
monitoring sites at longitudinal intervals (approximately 2.8°) of the model grids as shown by
gray rectangles (Fig. 4b) and calculated the long-term trend of averaged monitoring data at
each longitudinal band. The calculated increasing trends were clearly larger toward the west,
which was consistent with westward rise of the increasing trends of simulated data.
There were seasonal differences in the long-term increasing trend of surface $O_3$ over Japan.
The temporal variations of observed and simulated seasonal mean surface $O_3$ anomalies
during 1980–2005 averaged over Japan are shown in Fig. 5. The increasing trend of surface
$O_3$ over Japan in the monitoring data was greatest in spring (March–May: 4.04 ppbv/year) and
was also large in summer (June–August: 3.07 ppbv/year); in contrast, increasing trends were
relatively small in fall (September–November: 2.29 ppbv/year) and winter (December–
February: 1.28 ppbv/year). Seasonal dependency in the increasing trends of observed surface
$O_3$ over Japan has been previously reported (Ohara and Sakata, 2003; Naja and Akimoto,
2004; Parrish et al., 2012). Ohara and Sakata (2003) examined almost the same $O_3$ monitoring
data in Japan as used in the present study for the period 1985–1999 and showed year-round
increase in surface $O_3$ from 1985–1987 to 1997–1999 with a greater increase in the warm
season (March–August) than in the rest of the year. Naja and Akimoto (2004) also reported a
larger increase of $O_3$ in the warm season between the period 1970–1985 and 1986–2002 in the
boundary layer over Japan by analyzing ozonesonde data at four sites. Parrish et al. (2012)
summarized long-term changes in lower tropospheric baseline $O_3$ over the world including
two regions in Japan (Mt. Happo and several sites in the marine boundary layer grouped as
one region), and showed that the increasing trend of surface $O_3$ was greatest in spring and
least in fall in these two regions. In the present study, the simulated increasing trend in
seasonal mean surface $O_3$ was also larger in the warm (spring–summer) than in the cold
season (fall–winter), consistent with the observed increasing trends.
As described above, our model captured well the basic features of long-term trends in
observed surface $O_3$ over Japan, which allowed us to use the simulated data for further
analysis on the source of the long-term trend in the next section.

## 3.2 Contributions of $O_3$ production regions

The tracer-tagging simulation for 1980–2005 was conducted to examine the long-term
variations of $O_3$ tracers tagged by regions of photochemical production. IAVs in the annual





mean concentrations of each tagged $O_3$ tracer averaged over Japan are shown in Fig. 6. The
tagged tracers other than FT and stratosphere in Fig. 6 and the following figures represent the
contribution of $O_3$ produced in the PBL of different source regions shown in Fig. 1, where
contributions of several source regions were grouped into some combined source regions. It
should be noted that the model grids used for averaging in these figures differed from those in
Figs. 3–5. They encompassed almost all of Japan excluding the Nansei Islands in order to
examine temporal behavior of tagged $O_3$ tracers in all of Japan (see Fig. 4 for actual areas for
averaging).
Domestically created $O_3$ was the largest contribution to surface $O_3$ concentration averaged
over Japan during the whole simulation period. The contribution of domestic production had a
large IAV and was larger in the last decade than previously.
The second largest contribution was the $O_3$ created in the FT as a whole during almost the
entire period. For the FT, the northern mid-latitude regions such as North Pacific (NPC),
Europe (EUR), North Atlantic (NAT), North America (AMN), and China (CHN) made
leading contributions during the period; however, the increasing trend of these contributions
was considerable particularly for CHN and NPC (Fig. S2). Despite such differences among
the regional contributions in the FT, we hereafter only considered the total of each regional
contribution in the FT, since it was difficult to associate a regional contribution with a
particular source region of $O_3$ precursor emissions. The precursors eventually resulted in $O_3$
production in a region in the FT can be transported longer distance due to faster wind speed in
the FT and therefore would be influenced by emissions from a wider range of source regions
than in the PBL. The total FT contribution showed an increasing trend during the period.
The $NO_x$ emission from lightning was an indispensable source of $NO_x$ in the FT. The global
annual lightning-$NO_x$ emission in the current simulation was about 3.1 TgN/year averaged
over the entire period and showed a small but significant increase of about 0.012 TgN/year
(0.39 %/year). The increase in lightning-$NO_x$ emission was a consequence of changes in
convection activities due to the change in climate forced into the model during the period
(NCEP/NCAR meteorology and HadISST data). However, this increase in lightning-$NO_x$
emission was not the main cause of the increase in the contribution of the total FT – because a
sensitivity simulation with all emissions, $CH_4$ concentration, and stratospheric $O_3$ fixed at the
year 1980 level but with the same temporal evolution in climate showed a quite similar
increase in lightning-$NO_x$ emission but no significant increasing trend in the total FT
contribution. Therefore, the main cause of the increasing trend in the total FT contribution
was likely to be factors other than the increase in lightning-$NO_x$ emission.
The contribution of stratospheric $O_3$ was also large during the entire period, with
considerable temporal fluctuations. The large decreases of stratospheric contribution in the
early 1980s and 1990s stemmed from the decline of stratospheric $O_3$ concentration due to the
impact of large volcanic eruptions of Mt. El Chichon in 1982 and Mt. Pinatubo in 1991,
respectively (Akiyoshi et al., 2009).
In the early 1980s, the combined contributions of far remote regions from Japan in the
northern mid-latitude (Remote: EUR, NAT, and AMN) made a significant contribution, the
fourth largest, to the surface $O_3$ over Japan and remained at a steady level of contribution
during the study period. At the same time, the contribution of CHN significantly increased
from the mid-1980s, overtook the contribution of Remote in the early 1990s, and became the
largest single regional contribution – excluding the domestic one (i.e., JPN). Moreover, the
contributions of $O_3$ produced in the Korean Peninsula (KOR), the coastal regions in East Asia
[E-Asia-Seas: NPC, East China Sea (ECS), and Japan Sea (JPS)], and West-South-SouthEast



(WSSE) Asian regions [including Middle East (MES), India (IND), Indochina and Philippines
(IDC), and Indonesia etc. (IDN)] also showed obvious increasing trends.
The linear trend (ppbv/decade) of annual mean tagged $O_3$ tracers during the simulation
period as well as that of the total $O_3$, which is the sum of all tagged $O_3$ tracers averaged over
whole Japan (JPN-ALL) and those averaged over three sub-regions in Japan: western (JPN-
W), eastern (JPN-E), and northern (JPN-N) Japan is shown in Fig. 7 (see Fig. 4 for the
definition of sub-regions). The trend was calculated from the annual mean concentrations.
The increasing trend of total $O_3$ averaged over JPN-ALL was 2.37 ppbv/decade, which was
somewhat smaller than estimated in Fig. 3 (2.58 ppbv/decade) due to inclusion of model grids
in JPN-N for averaging where the simulated increasing trend of $O_3$ was relatively small. The
increasing trend of total $O_3$ tended to be greater westward. The absolute contribution of
domestically produced $O_3$ in Japan differed among the regions – it tended to be larger in JPN-
E than other parts of Japan (Nagashima et al., 2010); however, there were no such regional
differences in long-term trends. The westward tendency of larger increasing trends in total $O_3$
over Japan was mainly due to the similar tendency in the trends of the contribution of CHN,
KOR, and E-Asia-Seas, which strongly suggested a large impact of intra-regional
transboundary air pollution in East Asia. In particular, the increasing trend in the CHN
contribution was the largest for all sub-regions in Japan. The increasing trend in the
contributions of total FT and WSSE Asia was slightly smaller for JPN-N than for other parts
of Japan, which also contributed to the regional differences of the trend in total $O_3$ over Japan.
Interestingly, the contribution of Remote showed a small but significant increase only in JPN-
N – although emissions of $O_3$ precursors, $NO_x$ in particular, in Remote did not increase during
the period. Due to the large interannual fluctuation, the linear long-term trend of the
stratospheric contribution was non-significant for all regions in Japan.
The linear trend of tagged $O_3$ tracers and total $O_3$ averaged over all of Japan in spring,
summer, fall, and winter is shown in Fig. 8. The increasing trends of total $O_3$ in decreasing
order were spring, summer, winter, and fall. This is quite consistent with the seasonal
differences in the increasing trend of $O_3$ observed at several Japanese sites from the 1990s to
2011 (Parrish et al., 2012). The increasing trend in the CHN contribution was the largest of all
contributions in all four seasons and the trend was particularly large in spring. The KOR
contribution was also larger in spring than in other seasons, with the trend in summer of low
statistical significance due to relatively large IAVs. The contribution of E-Asia-Seas increased
significantly in all seasons. Seasonal differences in the increasing trend in the E-Asia-Seas
contribution were small, but were slightly larger in the warm (spring–summer) than the cold
season (fall–winter). The increasing trend in domestic (JPN) contribution was larger in spring
than in summer similarly to the cases of CHN and KOR contributions, but trends in both
seasons were non-significant; whereas those in the cold season were significantly larger than
in the warm season. The FT and WSSE Asian contributions showed semi-annual change in
their increasing trends; larger in summer and winter than in spring and fall. The contribution
of Remote showed a significant increasing trend only in winter; conversely that of Central-
North (CN) Asian regions [Central Asia (CAS) and East Siberia (ESB)] showed small but
significant decreasing trends in the cold season but non-significant trends in the warm season.
The seasonal features in each regional contribution described above enabled explanation of
the cause of the seasonality of increasing trend in total $O_3$ over Japan as follows. The largest
increasing trend of total $O_3$ in spring was predominantly attributed to the large increasing
trend in contributions of source regions in northeast Asia (CHN, KOR, E-Asia-Seas, and JPN).
The increasing trends in the contributions of CHN, KOR, and JPN were smaller in summer,
however, partly compensated by the growth of increasing trends in the FT and WSSE Asian





contributions from spring to summer. In the cold season, trends for most regions were smaller
than in the warm season, except for JPN. The increasing trend in contributions of northeast
Asian regions differed little between fall and winter; however, those of FT, WSSE Asia, and
Remote had larger increasing trends in winter than in fall, which made the increasing trend of
total $O_3$ in winter larger than in fall.
Table 1 summarizes the linear trends of annual mean tagged $O_3$ tracers and the total $O_3$
averaged over JPN-ALL. The vast majority (about 97 %) of the trend in total $O_3$ was balanced
with the sum of those trends in regional contributions with statistical significance. The largest
contribution was from the increase of $O_3$ produced in CHN (0.85 ppbv/decade), which
corresponded to about 36 % of the increasing trend of total $O_3$. The increasing trend in the
contribution of the total FT was also large (0.37 ppbv/decade), representing about 16 % of the
total $O_3$ trend. The contributions of northeast Asian regions other than CHN also increased
significantly (0.34, 0.29, and 0.27 ppbv/decade for KOR, E-Asia-Seas, and JPN, respectively)
and each accounted for about 12–15 % of the total $O_3$ trend. About 7 % of the total $O_3$ trend
was attributable to the increasing trend in WSSE Asian contributions (0.16 ppbv/decade). The
linear trends in the contributions of remaining regions [CN Asia, Remote, stratosphere, and
the others (OTH)] were small and non-significant, and so were not important concerning the
cause of reported surface $O_3$ increase over Japan.

**3.3  Impact of IAVs in $O_3$ precursor emissions in different source regions on**
**regional $O_3$ production**
The results in the preceding section revealed the relative importance of $O_3$ produced in
different regions to the recent increasing trend in surface $O_3$ over Japan. It is noteworthy that
this does not indicate the relative importance of the different regions of $O_3$ precursor
emissions. For example, there were significant contributions of E-Asia-Seas to the increasing
trend in surface $O_3$ over Japan, but there were clearly no large emission sources of precursors
in these maritime regions other than navigation. The increasing trend in the contribution of E-
Asia-Seas was likely a consequence of increased transport of $O_3$ precursors to this region,
which had been emitted in adjacent land areas. However, the tracer-tagging approach cannot
distinguish the differences in origins of emissions of precursors that resulted in $O_3$ production
in E-Asia-Seas. To further investigate the roles of different regions in the recent increasing
trend of surface $O_3$ over Japan, we performed a series of sensitivity simulations with different
assumptions for the temporal variation of factors, which would affect the surface $O_3$ over
Japan. Each sensitivity simulation consisted of a 26-year simulation with full-chemistry setup
of CHASER followed by another 26-year simulation with tracer-tagging setup of CHASER.
Initially, a sensitivity simulation was performed that was only forced by the IAVs in the
climate (NCEP/NCAR meteorology and HadISST data) but with all emissions of $O_3$
precursors, $CH_4$ concentration, and stratospheric $O_3$ fixed at the year 1980 level; then we
gradually added the increase or the IAV of chemical factors as summarized in Table 2. The
simulation F, driven by the IAV of all forcings, was identical to the standard simulation; and
simulation A was mentioned concerning lightning-$NO_x$ emission in the preceding section
500  (3.2).

The linear trends of annual mean total $O_3$ and tagged $O_3$ tracers that had significant effects
on the standard simulation averaged over all of Japan in all simulations are shown and
compared in Fig. 9. Simulation A showed no obvious increasing trend in total $O_3$ over Japan.



The JPN and total FT contributions exhibited increasing trends (0.12 and 0.06 ppbv/decade,
respectively), likely due to the IAV of the climate, but they were non-significant.
The increase in atmospheric concentration of $CH_4$ was added in simulation B, because this
would have a non-negligible impact on tropospheric $O_3$ (background $O_3$ in particular), as
frequently reported (Brasseur et al., 2006; Kawase et al., 2011; HTAP, 2010 and references
therein). In the simulations other than A, we used a $CH_4$ concentration increase rate of about
12.3 ppbv/year (0.73 %/year) during 1980–2000 and flattened thereafter. In simulation B, the
contribution of the total FT showed a significant increasing trend (0.18 ppbv/decade) as did
that of Remote (0.08 ppbv/decade; data not shown). The contributions of several other regions
such as CHN, E-Asia-Seas, and WSSE Asia also showed slight increasing trends
(approximately 0.01–0.02 ppbv/decade), although non-significant. Note that these values
included the impact of $CH_4$ increase as well as the IAV of the climate and, consequently, the
total $O_3$ in simulation B showed a significant increasing trend of about 0.44 ppbv/decade,
representing about 19 % of the increasing trend in total $O_3$ in the standard simulation (2.37
ppbv/decade).
In simulations C–E, the IAVs in emission of $O_3$ precursors in northeast Asian regions were
gradually added: CHN, KOR, and JPN, respectively. The increase in emissions of $O_3$
precursors in CHN in simulation C caused a large significant increasing trend in the
contribution of CHN itself (0.83 ppbv/decade). Moreover, the emission increase in CHN also
had a large impact on the contributions of other regions, in particular, the increase trends in
the contributions of KOR and E-Asia-Seas became significant: 0.12 and 0.15 ppbv/decade,
respectively. The JPN and the total FT contributions also showed somewhat larger increasing
trends in simulation C than in B, but the growth in trends between the two simulations was
not as large as those of KOR and E-Asia-Seas. The total effect of the emission increase in
CHN on the increasing trend in surface $O_3$ over Japan, assessed using the difference in total
$O_3$ trend between simulations B and C, was about 1.08 ppbv/decade and corresponded to
about 46 % of the increasing trend in total $O_3$ in the standard simulation. The relative
contribution of CHN as a source region of $O_3$ production to the surface $O_3$ increasing trend
over Japan was estimated as 36 % in the preceding section (3.2); however, the contribution of
CHN as a source region of $O_3$ precursors emission was somewhat (10 %) larger due to the
production of $O_3$ outside CHN. It is noteworthy that the slight increasing trend in the
contribution of WSSE Asia shown in the $CH_4$ increase in simulation B was smaller in
simulation C. The contributions of Remote and the stratosphere showed similar responses.
The increase in $O_3$ precursor emissions in CHN seemed to partly offset the increase in
influence of long range transport of $O_3$ from such regions.
The increase in emissions from KOR in addition to CHN in simulation D gave rise to a
much larger increasing trend in the contributions of KOR itself (0.38 ppbv/decade).
Compared with simulation C (0.12 ppbv/decade), about one-third of the increasing trend in
the contribution of KOR was attributed to the $O_3$ precursor emission increase in CHN and the
rest to emission increase in KOR. Similarly, the emission increase in KOR caused a larger
increasing trend in the contributions of E-Asia-Seas in simulation D (0.25 ppbv/decade). We
attributed about half of the increasing trend in the contribution of E-Asia-Seas in the standard
simulation (0.29 ppbv/decade) to the impact of $O_3$ precursor emission increase in CHN (and
partly that of the $CH_4$ increase: 0.15 ppbv/decade) as shown in simulation C, about one-third
to that in KOR, and the rest to that in regions other than northeast Asia. By further adding the
IAV in the domestic (JPN) emissions in simulation E, the increasing trend in the domestic
contribution became significant (0.28 ppbv/decade), implying that the increasing trend in
domestically produced $O_3$ was from a combination of multiple factors each of which did not





cause a significant increase. The total effect of the emission increase in KOR on the
increasing trend in surface $O_3$ over Japan assessed as the difference between simulations C
and D was about 0.38 ppbv/decade; and that of the IAV of domestic emission in Japan
assessed as the difference between simulations D and E was about 0.09 ppbv/decade; each of
which corresponded to about 16 and 4 % of the increasing trend in total $O_3$ in the standard
simulation, respectively.
The IAV in emissions of $O_3$ precursors in northeast Asian regions (CHN, KOR, and JPN)
together with the IAV in the climate and the increase in $CH_4$ concentration induced a
significant increasing trend in total $O_3$ over Japan with a rate of 1.99 ppbv/decade. This
accounted for about 84 % of the increasing trend in total $O_3$ in the standard simulation. The
rest of the increasing trend should be regarded as from $O_3$ precursor emission changes in
regions other than northeast Asia. The difference between simulations E and F (standard
simulation) showed that the emission change in such regions influenced surface $O_3$ over Japan
mainly through increasing the $O_3$ production in WSSE Asia and the FT (Fig. 9).

## 4   Summary and conclusion

We demonstrated the relative importance of the regions of photochemical $O_3$ production in
the global atmosphere on the long-term increasing trend in surface $O_3$ over Japan reported in
recent decades by conducting a series of tracer-tagging simulations using the global CTM
CHASER. The impact of the IAVs of climate, of $CH_4$ concentration, and of emission of $O_3$
precursors ($NO_x$ and NMVOC) in different source regions on regional photochemical $O_3$
production were also investigated.
The observed increasing trend of surface $O_3$ over Japan for 1980–2005 (2.70 ppbv/decade
for annual mean over whole Japan) was successfully reproduced by the model (2.58
ppbv/decade) including an obvious tendency of increase toward western Japan and to be
greater in the warm (spring–summer) than in the cold season (fall–winter).
The absolute contribution of each photochemical $O_3$ production region to the surface $O_3$ over
Japan represented by the concentrations of tagged $O_3$ tracer showed different temporal
evolution by regions. The contributions of all Asian regions except the northern part (i.e.,
CHN, KOR, E-Asia-Seas, JPN, and WSSE) as well as those of the total FT exhibited
significant increasing trends during the period. The increasing trend in the contribution of
domestically produced $O_3$ in Japan (i.e., JPN) did not differ much among the different regions
in Japan. However, there was a tendency in the increasing trends in contributions of CHN,
KOR, and E-Asia-Seas to be large toward western Japan, which was a main cause of the same
tendency in the increasing trend in total $O_3$ and suggested a large impact of intra-regional
transboundary air pollution in East Asia.
The trends in contributions of most $O_3$ production regions, except JPN, were larger in the
warm than in the cold season, providing a basis for the seasonality in the increasing trend in
total $O_3$ over Japan. Thus, the larger increasing trend of total $O_3$ in spring than in summer was
mainly due to the same tendency in increasing trends in the contributions of northeast Asian
regions (CHN, KOR, and JPN), although this was partly compensated by larger increasing
trends in the FT and WSSE Asia contributions in summer than spring. In the cold season, the
contributions of FT, WSSE Asia, and Remote had larger increasing trends in winter than in
fall, which led to a larger increasing trend in total $O_3$ in winter than in fall.
The sum of the trends in contributions of $O_3$ production regions with sufficient statistical
significance accounted for most (about 97 %) of the increasing trend in total $O_3$ over Japan

(2.37 ppbv/decade). The largest portion was attributed to the increasing trend of $O_3$ produced in CHN (36 %; 0.85 ppbv/decade), followed by that in the total FT (16 %; 0.37 ppbv/decade). The increasing trend in contributions of the other northeast Asian regions (KOR, E-Asia-Seas, and JPN; 0.27–0.34 ppbv/decade) each accounted for about 12–15 % of the total $O_3$ trend, and the majority of the rest of the total $O_3$ trend (7 %; 0.16 ppbv/decade) was attributable to WSSE Asia.

We further investigated the impact of the IAV of controlling factors, such as climate, $CH_4$ concentration, and emission of $O_3$ precursors, on photochemical $O_3$ production in different source regions and its influence on the long-term increasing trend in surface $O_3$ over Japan through a series of sensitivity simulations that gradually added the IAV of these factors. The IAV of the climate and the increase in $CH_4$ concentration together caused the increase of photochemical $O_3$ production in several regions and resulted in the significant increasing trend in surface $O_3$ over Japan (0.44 ppbv/decade) and represented about 19 % of the increasing trend in surface $O_3$ in the standard simulation. The increase in emission of $O_3$ precursors in CHN led to the increase of photochemical $O_3$ production in northeast Asian regions including CHN itself, KOR, JPN, and E-Asia-Seas; and the resulting increasing trend in surface $O_3$ over Japan (1.08 ppbv/decade) accounted for about 46 % of that in the standard simulation. The relative contribution of CHN to the surface $O_3$ increasing trend over Japan as the source region of $O_3$ precursor "emission" was 10 % larger than as the source region of $O_3$ "production" due to production of $O_3$ outside of CHN. Then, the impact of the $O_3$ precursor emission change in KOR and JPN on the increasing trend in surface $O_3$ over Japan (about 0.38 and 0.10 ppbv/decade, respectively) corresponded to 16 and 4 % of the increasing trend in total $O_3$ in the standard simulation, respectively. The rest of the increasing trend in total $O_3$ in the standard simulation (about 16 %) was attributed to $O_3$ precursor emission change in regions other than northeast Asia, mainly through increasing the photochemical $O_3$ production in WSSE Asia and the total FT.

*Acknowledgements.* This research was supported by the Global Environment Research Fund (S-7) by the Ministry of the Environment (MOE) of Japan and the East Asian Environment Research Program at the National Institute for Environmental Studies (NIES). We acknowledge the entire staff of the EANET and the AEROS air quality monitoring stations of the MOE of Japan and of the local governments for carrying out measurements and providing the observations. The calculations were performed on the NIES supercomputer system (NEC SX-8R, SX9). The GFD-DENNOU library was used for drawing the figures.



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





**Table 1.** Summary of the linear trends of annual mean tagged $O_3$ tracers as well as the total $O_3$
averaged over Japan (JPN-ALL) for 1980–2005. Bold figures denote that trends are
significant at 5 % risk level.

| Source Region | Trend [ppbv/dec] | Percent |
|:---:|:---:|:---:|
| CHN | **0.85 ± 0.2** | 35.8 |
| KOR | **0.34 ± 0.14** | 14.6 |
| JPN | **0.27 ± 0.19** | 11.5 |
| E-Asia-Seas | **0.29 ± 0.05** | 12.4 |
| WSSE Asia | **0.16 ± 0.04** | 6.8 |
| N Asia | -0.05 ± 0.08 | -2.1 |
| Remote | 0.04 ± 0.08 | 1.7 |
| OTH | 0.01 ± 0.02 | 0.5 |
| FT | **0.37 ± 0.1** | 15.5 |
| Strat. | 0.08 ± 0.28 | 3.3 |
| Total | **2.37 ± 0.42** | 100.0 |





**Table 2.** Summary of the sensitivity simulations and the standard simulation

| Simulation code | $CH_4$ concentration | $O_3$ precursor emissions | | | | Stratospheric $O_3$ trend |
|---|---|---|---|---|---|---|
| | | CHN | KOR | JPN | ROW[a] | |
| A | 1980[b] | 1980 | 1980 | 1980 | 1980 | 1980 |
| B | increase[c] | 1980 | 1980 | 1980 | 1980 | 1980 |
| C | increase | IAV[d] | 1980 | 1980 | 1980 | 1980 |
| D | increase | IAV | IAV | 1980 | 1980 | 1980 |
| E | increase | IAV | IAV | IAV | 1980 | 1980 |
| F (standard) | increase | IAV | IAV | IAV | IAV | IAV |

a Precursor emissions in the Rest Of the World (ROW) other than CHN, KOR, and JPN
b Each factor was fixed at the year 1980 level
c $CH_4$ concentration increased until 2000 and flattened thereafter
d InterAnnual Variation (IAV) of each factor was considered



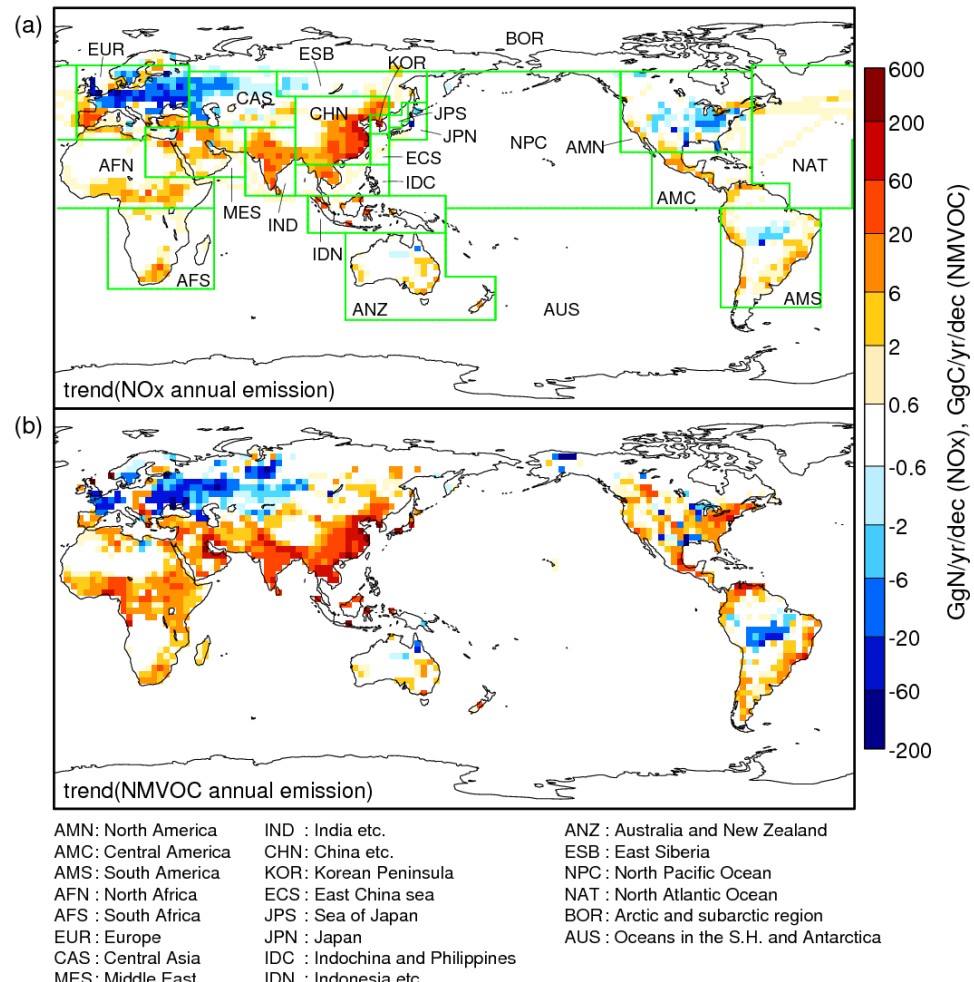

AMN: North America       IND : India etc.              ANZ : Australia and New Zealand
AMC: Central America      CHN: China etc.              ESB : East Siberia
AMS: South America       KOR: Korean Peninsula          NPC : North Pacific Ocean
AFN : North Africa        ECS : East China sea          NAT : North Atlantic Ocean
AFS : South Africa        JPS : Sea of Japan            BOR : Arctic and subarctic region
EUR : Europe            JPN : Japan                AUS : Oceans in the S.H. and Antarctica
CAS : Central Asia        IDC : Indochina and Philippines
MES : Middle East        IDN : Indonesia etc.


**Figure 1.** Linear trends of (a) NO$_x$ and (b) NMVOC emission during the simulation period (1980–2005) used in the study. Significant trends at 5 % risk level are colored. Source regions for tracer tagging are also displayed in the top figure.



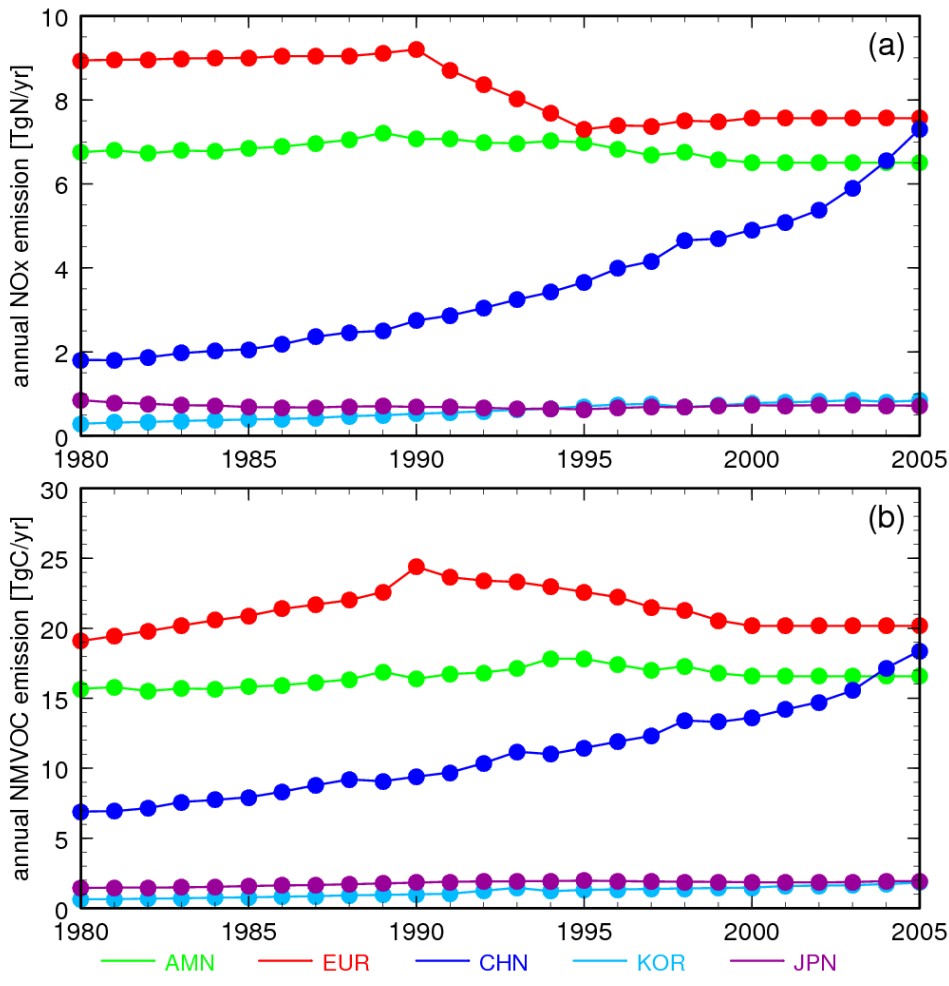

**Figure 2.** Temporal evolution of emissions of (a) $NO_x$ and (b) NMVOC averaged over several source areas in the Northern Hemisphere depicted in Fig. 1.



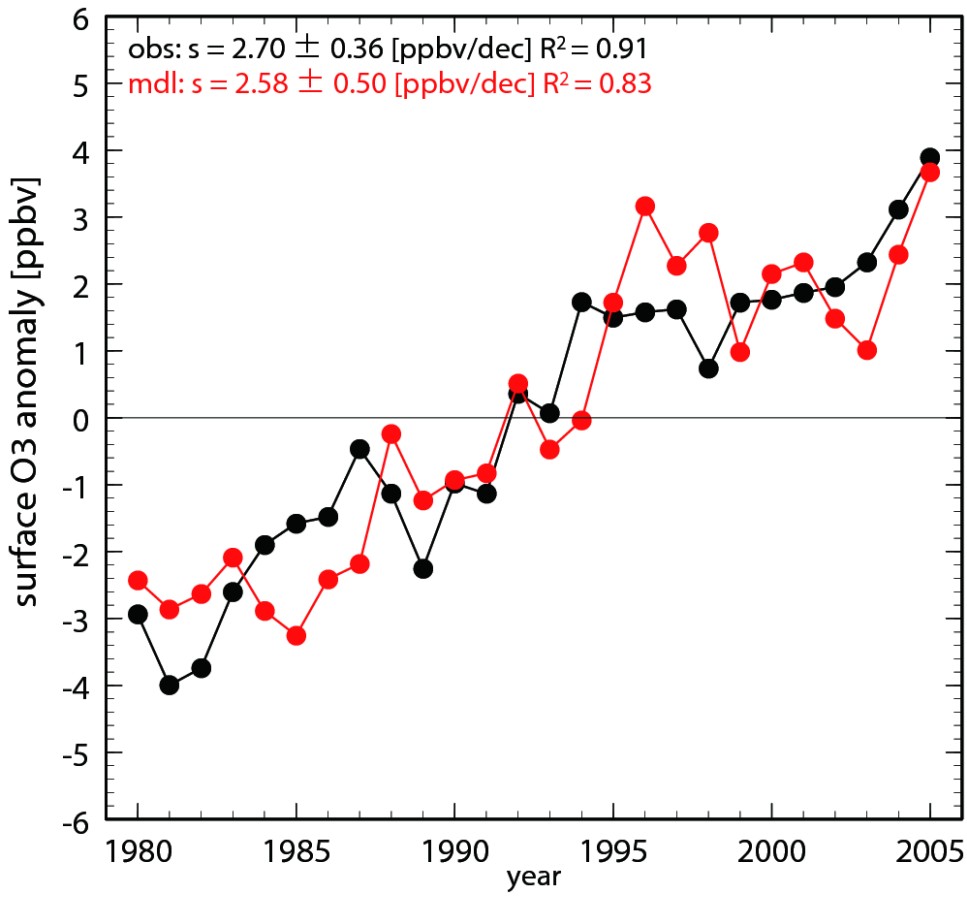

832

**Figure 3.** The temporal changes of annual mean surface O₃ anomaly averaged over Japan

from observation (AEROS: black) and model calculation (red). Anomalies are defined as

deviations from the values averaged over 1980–2005. The slope of a regression (s) for 1980–

2005 with their 95 % confidence interval and $R^2$ are also shown.




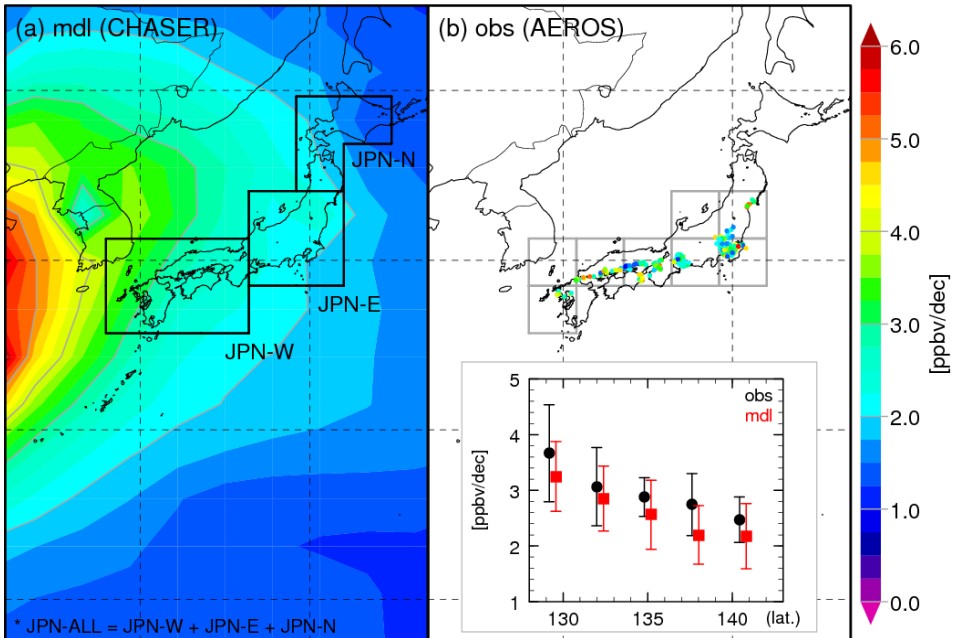

837

**Figure 4.** The linear trend of annual mean surface O$_3$ in 1980–2005 calculated from (a) model

simulations and (b) observations at AEROS monitoring sites. The inset in figure (b) shows the

longitudinal change of linear trends (black: AEROS observation; red: model) averaged within

the model grids shown by gray rectangles. The error bars denote their 95 % confidence

intervals. The black-rimmed areas in figure (a) are the area for averaging used in the figures

from Fig. 6. Note that JPN-ALL is the sum of JPN-W, JPN-E, and JPN-N areas and used for

the averaging in those figures.




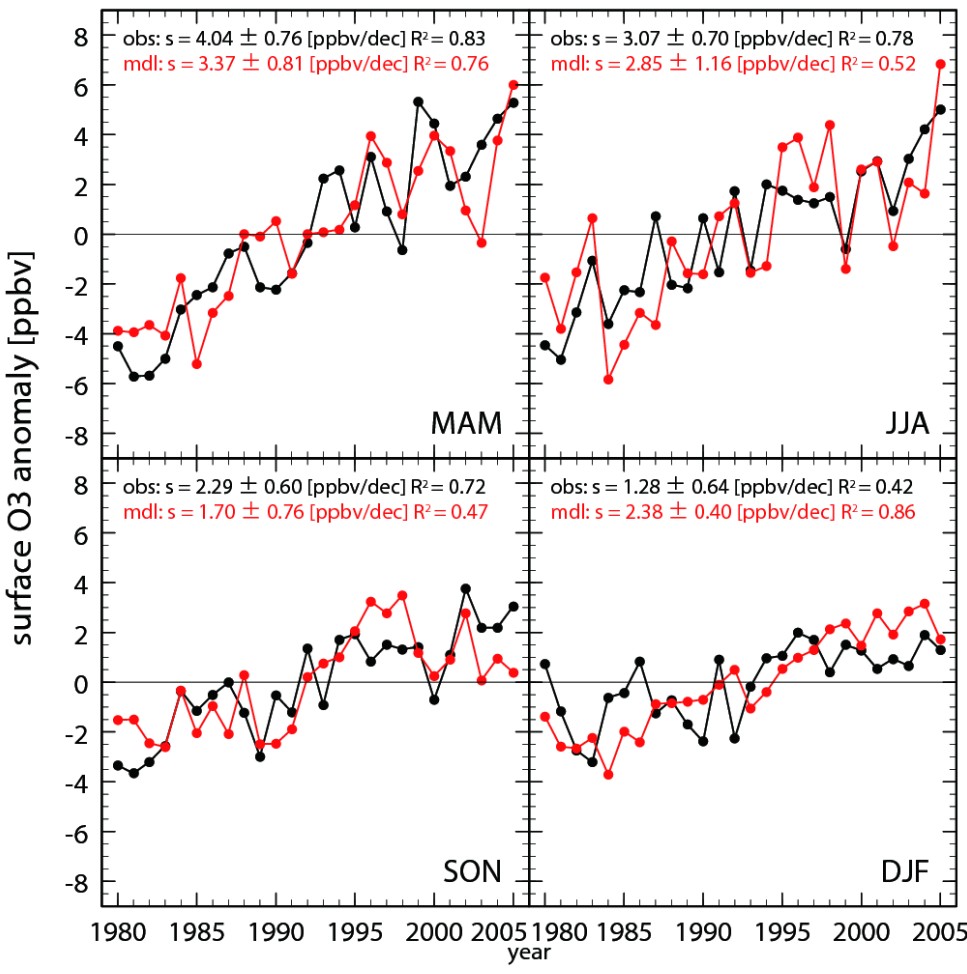

**Figure 5.** Same as Fig. 3 but the temporal changes of seasonal mean surface O₃ anomaly averaged over Japan from observations (AEROS: black) and model calculations (red).



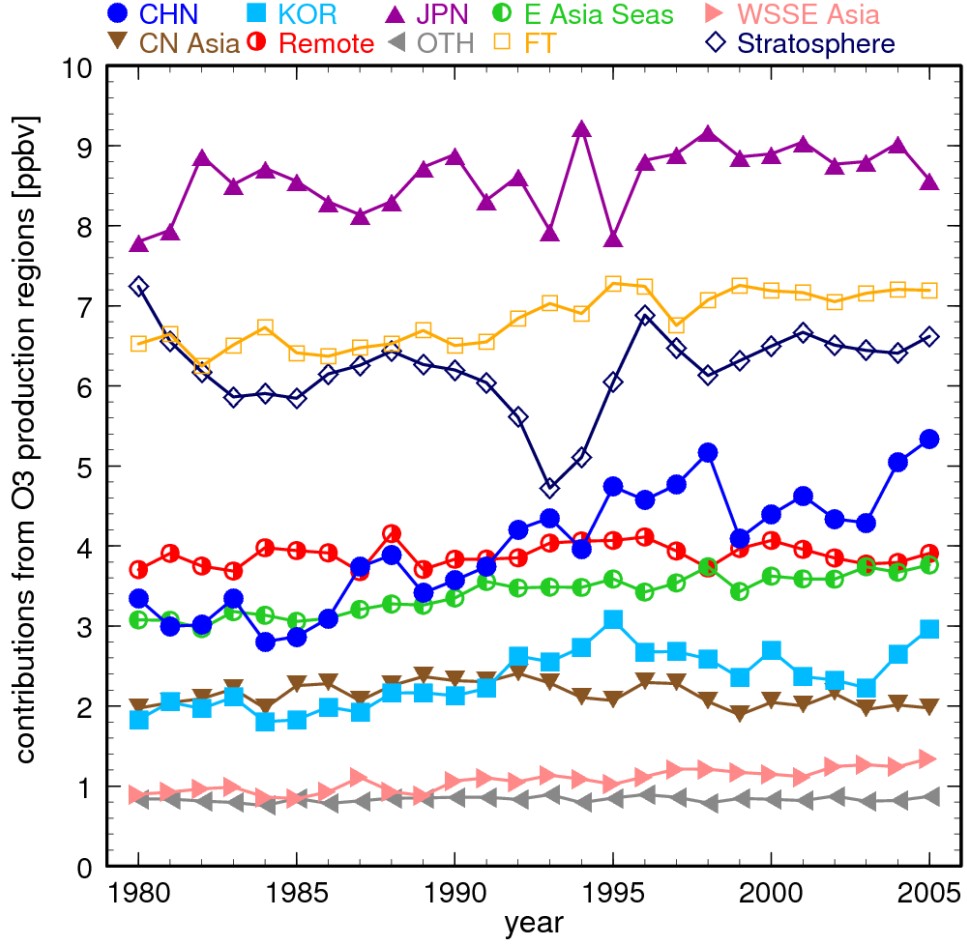

848

**Figure 6.** Long-term changes of annual mean contributions from source regions to surface O₃

over Japan. Some source regions are grouped: E-Asia-Seas is the sum of NPC, JPS, and ECS;

WSSE Asia is the sum of MES, IND, IDN, and IDC; CN Asia is the sum of CAS and ESB;

Remote is the sum of AMN, NAT, and EUR; and OTH is the other regions in the planetary

boundary layer.



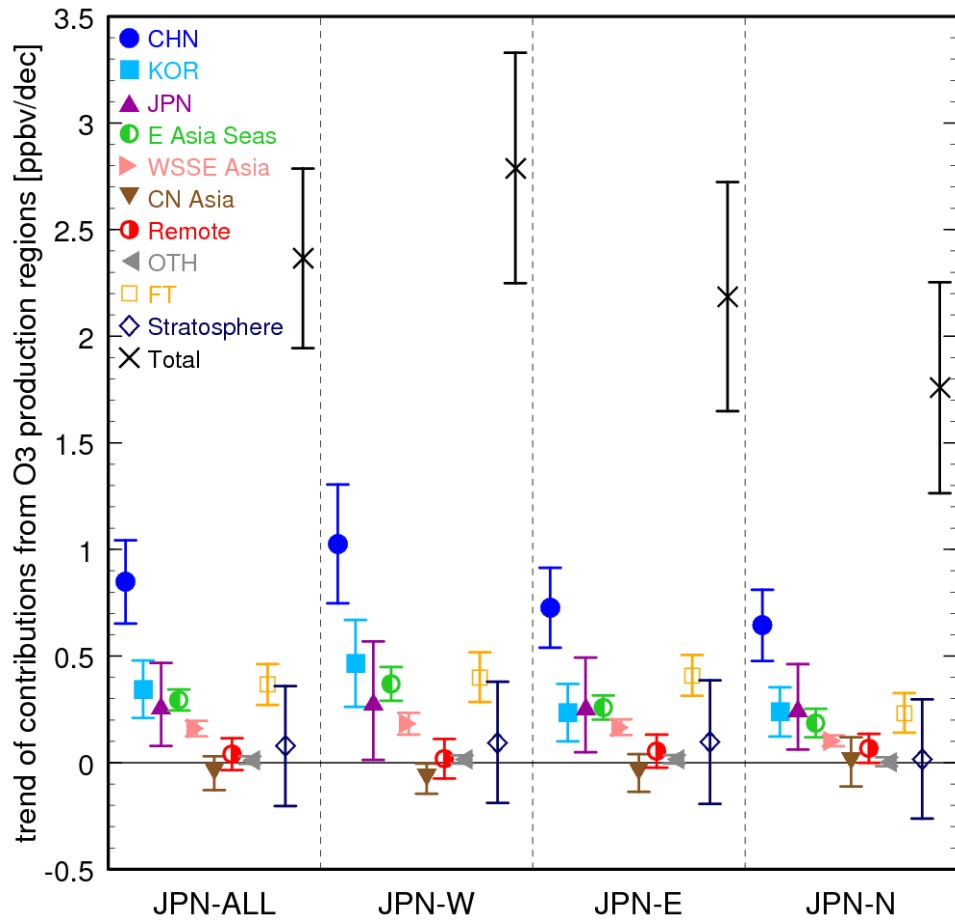


**Figure 7.** Linear trends of annual mean contributions in 1980–2005 from source regions to
surface O₃ over Japan shown in Fig. 6 (JPN-ALL) and those averaged in three sub-regions in
Japan (JPN-W, JPN-E, and JPN-N). Error bars are 95 % confidence intervals.





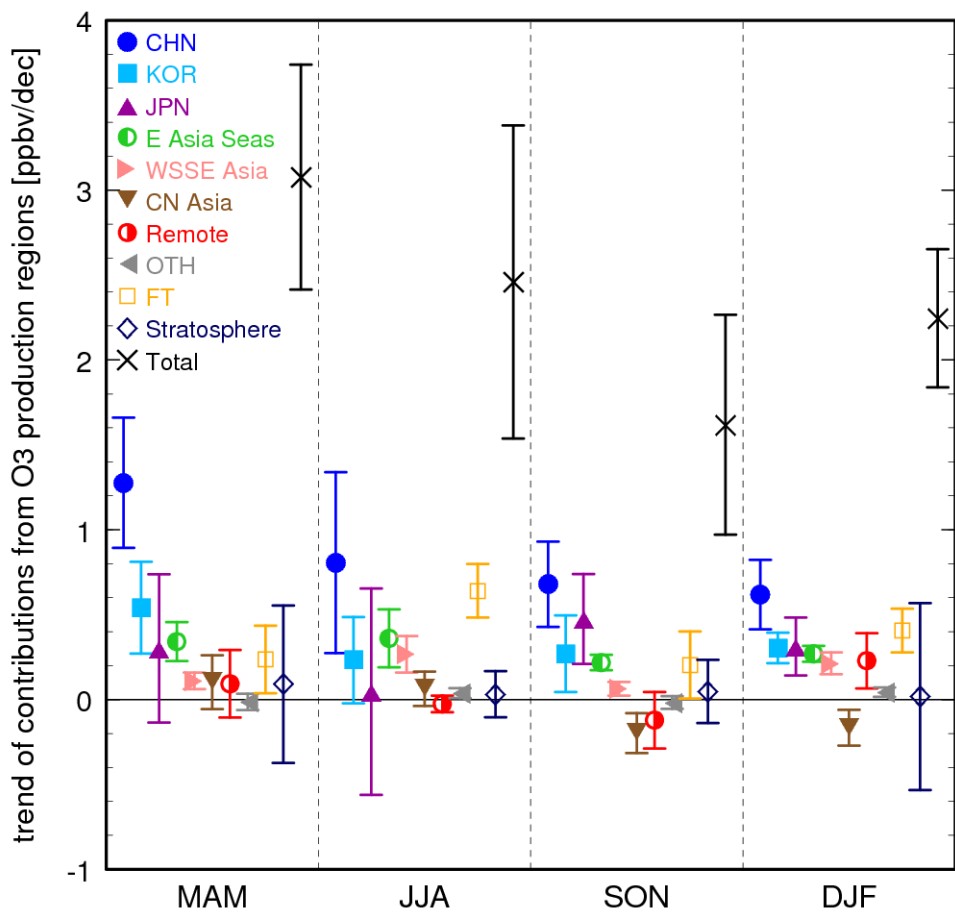


**Figure 8.** Linear trends of the contributions in 1980–2005 from source regions to surface $O_3$
over Japan in different seasons: spring (MAM), summer (JJA), fall (SON), and winter (DJF).
Error bars are 95 % confidence intervals.



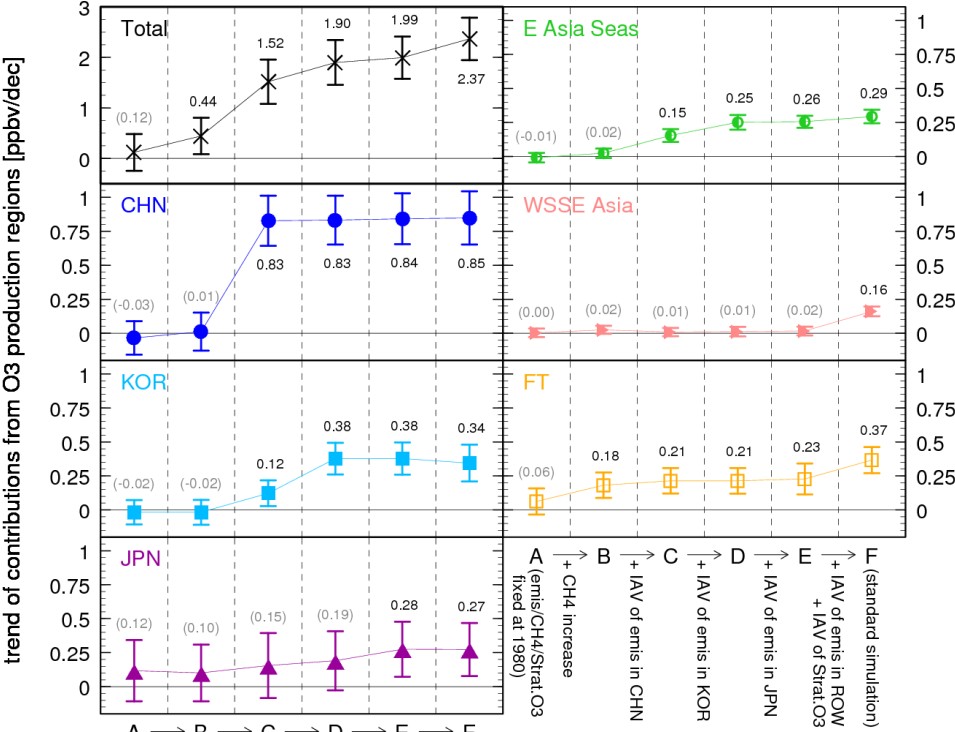

**Figure 9.** Linear trends of the annual mean contributions in 1980–2005 from source regions to surface O₃ over Japan in the sensitivity simulations and the standard simulation (error bars are 95 % confidence intervals). The exact values of the trends are also shown in the figure; the trends without sufficient statistical significance are shown in parentheses. The trends of each region's contribution in the simulations A–E and F (the standard simulation) are arranged from left to right in each panel.