# Peer review of "Long-term change in the contributions of various source"

_Atmospheric Chemistry and Physics, 2016_

## Referee Comment (RC1) · Anonymous Referee #1 · 20 Feb 2017

A review of "Long-term change in the contributions of various source regions to surface ozone over Japan" by Nagashima et al. submitted to ACP

General comments:

The authors made detailed tag-tracer simulations of tropospheric ozone and assessed the long-term changes in the source contributions to surface ozone in Japan. They also gathered currently available surface ozone data in Japan for the last 25 years (from 1980 through 2005) from the nationwide operational monitoring network by the Japanese EPA. The paper describes the model set-up including the inventories used and the tagging method, which is a key part, followed by the model results and discussions. The experiments were carefully made, and the interpretation was thoroughly detailed. This paper would be a nice piece of work contributing to the ozone commu-

nity. I only have several minor and/or technical comments that the authors can consider before publication, as listed below.

Major comments:

The paper is thoroughly written, but reads a bit too descriptive and too technically detailed. I would encourage the authors to make a bit more effort in reorganizing the sentences and try to put focus on the major scientific findings in this work.

Why up to 2005? and why surface ozone? Needs a bit of more sentences to justify these points.

Specific comments:

Title: I would prefer to "Long-term changes in the source contribution to surface ozone in Japan", just for your consideration.

Introduction, L49 - : The authors started mentioning the trends in Taiwan, China, and South Korea, but historically the trends over Japan were studied with ozone sondes or surface measurements prior to these areas. Hence, I would suggest the authors to start with Japanese trends then followed by recent reports in other countries.

P8, L318, section 3.3: The authors phrased "IAVs" in several places in the paper. The authors use the term "IAVs" not to mention (single) year-to-(single) year variability but rather decadal changes or changes during several years and the next several years (Explanation of Figure 3, for example). In Figure 3 the observed IAVs are not necessarily correlated with the modeled IAVs, on (single) year-by-year basis. So, I would encourage the authors to come back to this point and rephrase where necessary.

P7, L315-317; 2.70 and 2.58 ppbv/decade are too precise. I would suggest 2.7 and 2.6. But are these precise at 5% risk level?

P9, L374: last decade needs to be more specific. 2000s?

---

## Referee Comment (RC2) · Anonymous Referee #2 · 28 Feb 2017

This paper employed an global chemical transport model with on-line tracer-tagging method to investigate the long-term trends of surface ozone. This manuscript is interesting and belong to the scope of ACP.. This manuscript is well written. I suggested it can be published after considering the following comments.

General comment: 1. This study used the NCEP renalysis data to drive the Chaser model. Please compared the meteorological parameters with observations(surface or satellite) if possible. For example, cloud information and temperature.This is important to ozone simulation.

General comment 2: The author concluded that CHN contributed a lot to the trends of ozone in Japan. This can be expected because China's emissions are large and increases in last decades. I encourage the authors to analyze the contributing ability

of each regions to JPN ozone. For example, how many is the contribution of China per NOx/VOCs emissions increase to JPN O3 trends in unit: ppbv/Gg NOx or VOCs.

Major comment :

The authors should gave a short discussion on the uncertainties of models and its impact on the conclusions. for example, the emission inventory. REAS should be different with other inventories(MEIC or EDGAR4). I wonders if the difference between inventories affected the conclusions.

---

## Author Comment (AC1) · 5 May 2017

Response to the comment of Refree #1 The authors greatly appreciate your critical reading of our manuscript and highly valuable suggestions and comments. Our responses to your comments are listed below. (Pages and lines are those in the track-changed manuscript)

(RC): Refree Comment / (AR): Author Response

Major comments:

(RC) The paper is thoroughly written, but reads a bit too descriptive and too technically detailed. I would encourage the authors to make a bit more effort in reorganizing the sentences and try to put focus on the major scientific findings in this work.

[Figure]

(AR) Thank you for the overall comment. I read through the manuscript and checked the points which seems too descriptive and technically detailed, and delated or modified them if I could. The list of changes in the manuscript is below.

-P2, L39

-P4, L138-140 / L145-147

-P5, L180 / L183-184 / L204 / 206

-P6, L250 / L256-257

-P7, L304 / L306-307

-P8, L322-323 / L326-328 / L360-361 / L364-365

-P9, L366 / L378-379 / L407

-P10, L447-449 / L457

-P11, L458 / L466-468 / L490-491

-P12, L511

(RC) Why up to 2005? and why surface ozone? Needs a bit of more sentences to justify these points.

(AR1) The model simulation should include the period starting from 1980s and covering after the year 2000 to a certain extent to cover the years reported to have increasing trend in the surface O3 over Japan. Therefore, I selected the REAS v1.2 inventory because it was the only inventory data at the time of model calculation covering from 1980 to 2005 and focused on the whole East and Southeast Asian regions. The simulation period covering up to 2005 is mainly due to the temporal coverage of the emission inventory data used (REASv1.2). However, I'm strongly sure that I could obtain the basic understanding about the role of various source regions on the recent reported trend in surface O3 over Japan even with the simulation up to 2005 in this study, although

[Figure]

I know the anthropogenic emission of air pollutants in Asian region has been varied continuously even after 2005 and examining its impact on the air quality in Japan is also important. I added the following sentence to justify the reason more clearly (P4, L159-162).

"The end of simulation period (2005) was determined mainly due to the temporal coverage of the Asian emission data described below, however, this period sufficiently covered the years reported to have increasing trend in surface O3 over Japan in the previous literatures."

(AR2) The long-term increasing trend of the surface O3 in Japan during the last about 30 years despite of the continuous efforts to reduce the emission of O3 precursors in Japan and the consequent high violation rate of national ambient air quality standard (AAQS) in Japan, almost all the ambient air monitoring sites has been failed to meet the AAQS for a long time, are the persistent issues for environmental administration in Japan, therefore I focused on the surface O3. I added and modified the sentences stating the reason more clearly as follows (P2, L84 – P3, L92).

"In Japan, analysis of long-term observations by the ambient air quality monitoring network . . . until the present (Ohara and Sakata, 2003; . . . Akimoto et al., 2015). And the consequent high violation rate of national ambient air quality standard (AAQS) for surface O3 (hourly mean concentration of 60 ppbv) has been the persistent issue in environmental administration for a long time, therefore, there is an urgent need to study the reason for the increasing trend and examine the countermeasures. One clue is that the simultaneous observations of O3 precursors such as . . . inconsistent with the increasing trend of O3 over Japan."

Specific comments:

(RC) Title: I would prefer to "Long-term changes in the source contribution to surface ozone in Japan", just for your consideration.

[Figure]

(AR) Thank you for the suggestion. I also like the simple one. I change the title as you suggested.

(RC) Introduction, L49 - : The authors started mentioning the trends in Taiwan, China, and South Korea, but historically the trends over Japan were studied with ozone sondes or surface measurements prior to these areas. Hence, I would suggest the authors to start with Japanese trends then followed by recent reports in other countries.

(AR) Thank you for the suggestion. I changed the manuscript accordingly as follows. (P2, L56-83)

"Japan experienced a rapid industrialization ahead of other Asian countries, and an increasing trend has been found in various observations of tropospheric O3 . . . until the mid-2000s (Tanimoto, 2009; Tanimoto et al., 2009; Parrish et al., 2012). During the recent decades, an increasing trend in tropospheric O3 has also been observed at . . . in tropospheric O3 for other regions in the world (Cooper et al., 2014).

(RC) P8, L318, section 3.3: The authors phrased "IAVs" in several places in the paper. The authors use the term "IAVs" not to mention (single) year-to-(single) year variability but rather decadal changes or changes during several years and the next several years (Explanation of Figure 3, for example). In Figure 3 the observed IAVs are not necessarily correlated with the modeled IAVs, on (single) year-by-year basis. So, I would encourage the authors to come back to this point and rephrase where necessary.

(AR) I carefully checked all the "IAVs" and rephrased some of them which do not mean year-by-year variation but rather longer (e.g. decadal) temporal variation to "long-term variation" or "temporal variation". The following is the list of changes.

-P8, L334: "IVAs -> long-term variation"

-P11, L494 (title of the section 3.3): "IVAs -> temporal variations"

-P12, L513: "increase or the IAV -> temporal variation"

-P12, L514: "IVA -> temporal variation"

-P12, L542: "IVAs -> temporal variations"

-P13, L572: "IVA -> temporal variation"

-P14, L627 / L631: "IVA -> temporal variation"

-Table 2: "IAV -> Var", "InterAnnual Variation (IAV) -> Temporal Variation (Var)"

(RC) P7, L315-317; 2.70 and 2.58 ppbv/decade are too precise. I would suggest 2.7 and 2.6. But are these precise at 5% risk level?

(AR) Yes. These trend values are precise at 5 % risk level. So, I'll keep them as is.

(RC) P9, L374: last decade needs to be more specific. 2000s?

(AR) It means the period from 1996 to 2005. I added the period in the manuscript as follows. (P9, L390)

"The contribution of domestic production had a large IAV and was larger in the last decade (1996-2005) than previously."

Best regards,

Tatsuya Nagashima

Please also note the supplement to this comment:
http://www.atmos-chem-phys-discuss.net/acp-2016-1087/acp-2016-1087-AC1-supplement.pdf
* * *

---

## Author Comment (AC2) · 5 May 2017

Response to the comment of Referee #2

The authors greatly appreciate your critical reading of our manuscript and highly valuable suggestions and comments. Our responses to your comments are listed below. (Pages and lines are those in the track-changed manuscript)

(RC): Referee Comment / (AR): Author Response

General comment 1:

(RC) This study used the NCEP renalysis data to drive the Chaser model. Please compared the meteorological parameters with observations (surface or satellite) if possible. For example, cloud information and temperature. This is important to ozone simulation.

(AR) Thank you for the comment. Unfortunately, I didn't output any cloud parameters simulated in the model, but I could compare the surface temperature used in the model with those observed in Japan which were compiled by Japan Meteorological Agency (JMA). JMA selected 15 sites which undergo little urban influences to derive the average surface temperature over Japan. The modelled annual mean surface temperature averaged over whole Japan showed a significant warming during the simulation period, $0.44 \pm 0.21$ °C/decade, which well matched the observed warming of $0.45 \pm 0.23$ °C/decade. Add to this long-term trend, the inter-annual (year-to-year) variation was also well captured by the model, although there was a discrepancy that the modelled temperature was somewhat warmer than the observation in 2000s particularly in winter which might be related to the slight overestimation of winter surface O3 in the model depicted in the Fig.5. I added the following sentences to the manuscript. (P12, L520-526)

"The surface temperature over Japan in the model which was assimilated into NCEP/NCAR reanalysis data showed a warming of $0.44 \pm 0.21$ °C/decade in the annual mean during the simulation period which well corresponded to the observed warming of $0.45 \pm 0.23$ °C/decade (JMA, 2017). The IAV of the surface temperature was well captured by the model too, although the modelled temperature was somewhat warmer than the observation in 2000s particularly in winter which might be related to the slight overestimation of winter surface O3 in the model depicted in Fig.5."

General comment 2:

(RC) The author concluded that CHN contributed a lot to the trends of ozone in Japan. This can be expected because China's emissions are large and increases in last decades. I encourage the authors to analyze the contributing ability of each regions to JPN ozone. For example, how many is the contribution of China per NOx/VOCs emissions increase to JPN O3 trends in unit: ppbv/Gg NOx or VOCs.

(AR) Thank you for the comment, the concept of the contributing ability is quite interesting. However, because the sensitivity simulations to derive the contribution of emission trend in each region was done by varying all the O3 precursors emission simultaneously, it is not straightforward to estimate the contributing ability of a single O3 precursor individually. Nevertheless, I tried to estimate the contributing ability by conducting a multiple regression analysis with NOx and VOC emissions in a source region (e.g CHN) as explanatory variables and the contribution of that region on the surface O3 over Japan as the target variable, however, the analysis was failed because of the high correlation between NOx and VOC emission trends in the source regions such as CHN (p-value of the regression coefficient for NOx is 0.18) and KOR (that for VOC is 0.17). Therefore, deriving the contributing ability of different source regions should be addressed in the future study with a careful experimental design for that purpose.

Major comment:

(RC) The authors should gave a short discussion on the uncertainties of models and its impact on the conclusions. for example, the emission inventory. REAS should be different with other inventories (MEIC or EDGAR4). I wonders if the difference between inventories affected the conclusions.

(AR) I added a short discussion on the uncertainty of O3 precursor emission inventories and possible impacts on the conclusions in the last chapter as follows. (P14, L647-658)

"The results summarized above depended largely on the forcings of long-term simulation, particularly the long-term variation of the emissions of O3 precursors in Asia. Zhao et al. (2013) estimated the NOx emission in China for the period 1995—2010 and compared it to the existing emission inventories including Hao et al. (2002), Zhan et al. (2007), and the version of REAS used in this study. They showed the log-term increasing trend in Chinese NOx emission in REAS was consistent with that in the other inventories, but the amount of emission was somewhat smaller in REAS than in the others. Therefore, the long-term increasing trend in the contribution of Chinese emission to the surface O3 over Japan showed in the preset study would be retained if the other emission inventories were used for the simulation but the specific values of the contributions could be affected. Further studies should address the impact these uncertainties in the different emission inventories on the trend of surface O3 over Japan."

Best regards,

Tatsuya Nagashima

Please also note the supplement to this comment:
http://www.atmos-chem-phys-discuss.net/acp-2016-1087/acp-2016-1087-AC2-supplement.pdf
* * *
[Figure]

**Supplement:**

[revised manuscript text omitted]

AMN: North America
AMC: Central America
AMS: South America
AFN : North Africa
AFS : South Africa
EUR : Europe
CAS : Central Asia
MES : Middle East

IND : India etc.
CHN: China etc.
KOR: Korean Peninsula
ECS : East China sea
JPS : Sea of Japan
JPN : Japan
IDC : Indochina and Philippines
IDN : Indonesia etc.

ANZ : Australia and New Zealand
ESB : East Siberia
NPC : North Pacific Ocean
NAT : North Atlantic Ocean
BOR: Arctic and subarctic region
AUS : Oceans in the S.H. and Antarctica

**Figure 1.** Linear trends of (a) $NO_x$ and (b) NMVOC emission during the simulation period (1980–2005) used in the study. Significant trends at 5 % risk level are colored. Source regions for tracer tagging are also displayed in the top figure.

[Figure]

**Figure 2.** Temporal evolution of emissions of (a) NO$_x$ and (b) NMVOC averaged over several source areas in the Northern Hemisphere depicted in Fig. 1.

[Figure]

**Figure 3.** The temporal changes of annual mean surface O₃ anomaly averaged over Japan from observation (AEROS: black) and model calculation (red). Anomalies are defined as deviations from the values averaged over 1980–2005. The slope of a regression (s) for 1980–

2005 with their 95 % confidence interval and $R^2$ are also shown.

[Figure]

**Figure 4.** The linear trend of annual mean surface $O_3$ in 1980–2005 calculated from (a) model simulations and (b) observations at AEROS monitoring sites. The inset in figure (b) shows the longitudinal change of linear trends (black: AEROS observation; red: model) averaged within the model grids shown by gray rectangles. The error bars denote their 95 % confidence intervals. The black-rimmed areas in figure (a) are the area for averaging used in the figures from Fig. 6. Note that JPN-ALL is the sum of JPN-W, JPN-E, and JPN-N areas and used for the averaging in those figures.

[Figure]

**Figure 5.** Same as Fig. 3 but the temporal changes of seasonal mean surface $O_3$ anomaly averaged over Japan from observations (AEROS: black) and model calculations (red).

[Figure]

**Figure 6.** Long-term changes of annual mean contributions from source regions to surface $O_3$ over Japan. Some source regions are grouped: E-Asia-Seas is the sum of NPC, JPS, and ECS; WSSE Asia is the sum of MES, IND, IDN, and IDC; CN Asia is the sum of CAS and ESB; Remote is the sum of AMN, NAT, and EUR; and OTH is the other regions in the planetary boundary layer.

[Figure]

**Figure 7.** Linear trends of annual mean contributions in 1980–2005 from source regions to surface $O_3$ over Japan shown in Fig. 6 (JPN-ALL) and those averaged in three sub-regions in Japan (JPN-W, JPN-E, and JPN-N). Error bars are 95 % confidence intervals.

[Figure]

**Figure 8.** Linear trends of the contributions in 1980–2005 from source regions to surface O$_3$

over Japan in different seasons: spring (MAM), summer (JJA), fall (SON), and winter (DJF).

Error bars are 95 % confidence intervals.

[Figure]

**Figure 9.** Linear trends of the annual mean contributions in 1980–2005 from source regions to surface O₃ over Japan in the sensitivity simulations and the standard simulation (error bars are 95 % confidence intervals). The exact values of the trends are also shown in the figure; the trends without sufficient statistical significance are shown in parentheses. The trends of each region's contribution in the simulations A–E and F (the standard simulation) are arranged from left to right in each panel.